# Updated MISR Over-Water Research Aerosol Retrieval Algorithm Part 2: A Multi-Angle Aerosol Retrieval Algorithm for Shallow, Turbid, Oligotrophic, and Eutrophic Waters

James A. Limbacher[1,2,3] and Ralph A. Kahn[1]

[1]Earth Science Division, NASA Goddard Space Flight Center, Greenbelt, 20771, USA
[2]Science Systems and Applications Inc., Lanham, 20706, USA
[3]Department of Meteorology and Atmospheric Science, The Pennsylvania State University, State College, 16802, USA

*Correspondence to*: James A. Limbacher (James.Limbacher@nasa.gov)

**Abstract.** Coastal waters serve as transport pathways to the ocean for all agricultural and other runoff from terrestrial sources, and many are the sites for upwelling of nutrient-rich, deep water; they are also some of the most biologically productive on Earth. Estimating the impact coastal waters have on the global carbon budget requires relating satellite-based remote-sensing retrievals of biological productivity (e.g., Chlorophyll-a concentration) to in-situ measurements taken in near-surface waters. The Multi-angle Imaging SpectroRadiometer (MISR) can uniquely constrain the "atmospheric

correction" needed to derive ocean color from remote-sensing imagers. Here, we retrieve aerosol amount and type from MISR over all types of water. The primary limitation is an upper bound on aerosol optical depth (AOD), as the algorithm must be able to distinguish the surface. This updated MISR research aerosol retrieval algorithm (RA) also assumes that light reflection by the underlying ocean surface is Lambertian. The RA computes the ocean surface reflectance ($R_{rs}$) analytically for a given AOD, aerosol optical model, and wind speed.

We provide retrieval examples over shallow, turbid, and eutrophic waters and introduce a productivity/turbidity index (PTI), calculated from retrieved spectral $R_{rs}$, that distinguished water types (similar to NDVI over land). We also validate the new algorithm by comparing spectral AOD and Ångström exponent (ANG) results with 2419 collocated AERosol RObotic NETwork (AERONET) observations. For AERONET 558 nm interpolated AOD < 1.0, the root-mean-squared-error (RMSE) is 0.04 and linear correlation coefficient is 0.95. For the 502 cloud-free MISR/AERONET collocations with an

AERONET AOD > 0.20, the ANG RMSE is 0.25 and r=0.89. Although MISR RA AOD retrieval quality does not appear to be substantially impacted by the presence of turbid water, MISR RA-retrieved Ångström exponent seems to suffer from increased uncertainty under such conditions.

MISR supplements current ocean color sources in regions where sun glint precludes retrievals from single-view-angle instruments. MISR atmospheric correction should also be more robust than that derived from single-view instruments such

as MODIS. This is especially true in regions of shallow, turbid, and eutrophic waters, locations where biological productivity can be high, and single-view angle retrieval algorithms struggle to separate atmospheric from oceanic features.

# 1 Introduction

Although coastal and generally turbid waters represent some of the most biologically productive waters in the world, constraining their impacts on the global carbon budget remains challenging (*Bauer et al.*, 2013; *Behernfeld et al.,* 2005).

One reason is the need to account for the contribution the atmosphere makes to the observed, top-of-atmosphere (TOA) signal obtained from frequent, large-scale, remote-sensing observations. Aerosol loading tends to be considerably higher over coastal waters than over the remote ocean, and the inability of many aerosol retrieval algorithms to separate surface from atmospheric features in complex coastal water zones precludes accurate surface retrievals.

The NASA Earth Observing System's (EOS) Multi-angle Imaging SpectroRadiometer (MISR) instrument was launched aboard the Terra satellite on December 18th, 1999. The satellite moves north to south on the day side of the planet, in a ~700 km sun-synchronous orbit with a local 10:30 AM equator crossing time (dayside) and a ~98° orbital inclination. Originally intended as a six-year mission, MISR continues to operate nominally more than 18 years later. The instrument samples upwelling radiance over a ~400 km wide swath in four spectral bands centered at 446 nm (blue), 558 nm (green), 672 nm

(red), and 866 nm (near-infrared, or NIR) (*Diner et al.,* 1998). These measurements are taken for each of the nine MISR cameras, viewing in both the forward and aft directions along the satellite's orbit ($\pm$ 70.5°, 60.0°, 45.6°, 26.1°, and 0.0° or nadir), resulting in an optical path length that varies from one to three. (Throughout this paper, the nadir camera is designated "An," the forward and aft-viewing 26.1˚ cameras "Af" and "Aa," respectively, and those viewing at 45.6°, 60.0°, and 70.5° as "B," "C," and "D," respectively.) MISR radiance data are generally recorded at 1.1 km resolution, but data for

all nine cameras from the red band are retained at 275m, as are data from all four bands at nadir. Additionally, MISR samples selected regions at roughly 275 m in all 36 channels (the A, B, C, and D cameras have different focal lengths to provide uniform spatial resolution) consistently for ~15 years (i.e., consistent, long-term, high-resolution data), resulting in a high-resolution, high-information-content dataset suitable for heterogeneous coastal water retrievals.

In recent years, many in-situ measurements of near-surface Chlorophyll-a concentration (*Chl*, and other biological constituents) have been taken jointly with measurements of remote-sensing reflectance ($R_{rs}$; $sr^{-1}$). $R_{rs}$ represents the *normalized* (see *Morel et al.*, 2002) upward directed radiance ($L_w$; W $m^{-2}$ $\mu m^{-1}$ $sr^{-1}$) just above the ocean surface, divided by the bottom-of-atmosphere, downward-directed spectral irradiance ($E_d$; W $m^{-2}$ $\mu m^{-1}$). $R_{rs}$ is widely used in the ocean color community because it can be directly related to near-surface biological proxies (measured in-situ; such as Chlorophyll-a),

and can be estimated in the field from surface remote-sensing observations, as described in *Mobley* [1999]. As much remote-sensing-based ocean color research has focused on relating measurements of biological proxies to $R_{rs}$ empirically (*Werdell and Bailey*, 2002; 2005), two different strategies have emerged that make use of $R_{rs}$. The purely empirical

approach to retrieving ocean color properties relates ratios of normalized $R_{rs}$ to parameters such as *Chl* statistically (e.g. see *O'Reilly et al., 1998, Morel and Gentili*, 2009, *Hu et al.*, 2012). A semi-analytical method involves using a combination of radiative transfer theory (*Morel et al.,* 2002) and empirical observations (e.g. *Morel and Prieur*, 1977, *Lee et al.*, 2015) to retrieve *Chl* and other parameters (e.g. *Maritorena et al., 2002*, *Werdell et al.*, 2013). Fundamentally, both techniques rely on high-quality retrievals of $R_{rs}$, which is dependent on the quality and spectral dependence of the atmospheric correction (*Kahn et al.,* 2016), as well as the calibration of the instrument.

*Mobley et al.* [2016] summarize the atmospheric correction procedure implemented in the ocean color algorithm used for the MODerate resolution Imaging Spectroradiometer (MODIS) and the Sea-Viewing Wide Field-of-View Sensor (SeaWiFS) at NASA Goddard Space Flight Center (GSFC). For dark (Case I) water retrievals, where the remote-sensing reflectance is negligible for the near-infrared wavelengths (NIR), the retrieval process follows directly from the one proposed by *Gordon and Wang* [1994], with updates made to the aerosol optical models (*Ahmad et al.*, 2010) and spectral bands used. For Case II waters, where $R_{rs}$ in two NIR bands used is not necessarily negligible (or even identical), the algorithm follows an iterative scheme described in *Bailey et al.* [2010], initialized with a guess of $R_{rs}$(NIR)=0.

With nine cameras taking measurements at four wavelengths, MISR has the angular information content needed to constrain aerosol properties, even over bright turbid water, but lacks the specific spectral bands for ocean color applications, though it can fill in glint-contaminated regions of single-view instruments such as MODIS (*Limbacher and Kahn*, 2017). As such, this paper focuses on MISR's ability to retrieve the atmospheric component, aerosol amount and type, over shallow, turbid, and eutrophic waters, as well as over oligotrophic waters. The paper is organized as follows: section 2 outlines the datasets and methodologies used, example retrievals are shown in section 3, validation is presented in section 4, and conclusions are given in section 5.

## 2 Upgraded MISR Research Aerosol Retrieval Algorithm (RA) and Comparison datasets

### 2.1 The Over-Water MISR RA Methodology

The MISR standard aerosol product (SA; Diner et al., 2008; Martonchik et al., 2009) provides publicly accessible aerosol amount and type information globally at 4.4 km spatial resolution (Kahn et al., 2010, Kahn and Gaitley, 2015; Garay et al., 2017). In contrast, the RA can only process MISR data for selected locations and times, on a case-by-case basis, but offers spatial resolution down to 1.1 km or 275 m pixel size, advancements in radiometric calibration critical for aerosol-type retrieval, an improved surface representation, and the option of a greatly expanded aerosol optical model climatology (Limbacher and Kahn, 2014; 2015; 2017). Briefly, the RA finds the set of aerosol optical models, associated aerosol amounts, and *Chl* values, that minimize the difference between the observed TOA reflectances and simulated values

that are stored in a look-up table (LUT). These dimensionless TOA reflectances are computed from the observed radiances according to the following:

$$\rho_{\lambda,c}^{TOA} = L_{\lambda,c} * \frac{\pi * D^2}{E_\lambda^{TOA}} \tag{1}$$

where $L_{\lambda,c}$ represents the TOA observed radiance (W m$^{-2}$ μm$^{-1-1}$ sr$^{-1}$) in band $\lambda$ and camera $c$, D is the Earth-Sun distance at the time of observation in Astronomical Units (AU), and $E_\lambda^{TOA}$ is the exo-atmospheric solar irradiance at 1 AU (W m$^{-2}$ μm$^{-1}$). We then correct these TOA reflectances for the following (described in *Limbacher and Kahn* [2015; 2017]): gas absorption, out-of-band light, stray-light from instrumental artifacts, flat-fielding (pixel-to-pixel inconsistency), and temporal calibration trends. Even with the near-nadir-camera ghosting corrections implemented in *Limbacher and Kahn* [2015], we found it necessary to include an additional veiling-light correction (similar to *Witek et al.,* [2017]) to bring the AODs into better agreement with AERONET; ghosting corrections for the off-nadir cameras are currently under study.

The fundamental physical assumption of this algorithm is that we *assume* the *camera-averaged* remote-sensing reflectances (R$_{rs}$; including any reflection off of the underlying sea-floor) can be adequately approximated as Lambertian. (This is supported by the f/Q corrections found in *Morel et al.*, 2002). The addition of an f/Q correction (f/Q represents a non-Lambertian bi-directional surface modification; such as from *Morel et al.*, 2002) would increase the robustness of this algorithm over deep-water regions, at the expense of computational time and simplicity. We chose to keep the model simple for this manuscript. The objective of our algorithm is to self-consistently retrieve AOD and remote-sensing reflectance at 1.1 km resolution, for any set of aerosol optical models used, such as those found in *Limbacher and Kahn* [2014]. We run the RA for all optical models found in our LUT, aggregating the resulting AOD and aerosol properties corresponding to the best-fitting AOD for each candidate aerosol mixture according to an exponential weighting of our cost function (*M*),

$$M = \sum_\lambda \sum_c \frac{w_{\lambda,c} * \left[\rho_{\lambda,c}^{TOA} - (\rho_{\lambda,c}^{Model} + E_\lambda^{BOA} * R_{rs,\lambda} * T_{\lambda,c}^{up})\right]^2}{Unc_{\lambda,c}^2 * \left[\sum_\lambda \sum_c w_{\lambda,c}\right]}, \tag{2a}$$

$$Mix\_Weight = exp\left(\frac{M_{Min} - M}{M_{Min} + 0.01}\right). \tag{2b}$$

The channel-specific weights are $w_{\lambda,c}$ (detailed in Section 2.1.3 below), the assumed uncertainty for the model-measurement system is $Unc_{\lambda,c}$ (detailed in 2.1.3), and $\rho_{\lambda,c}^{Model}$ represents the effects of atmospheric gas and aerosol scattering, plus the combined TOA effects of Fresnel reflection (modeled as an isotropic Cox-Munk surface) and whitecaps. The term $E_\lambda^{BOA} * R_{rs,\lambda} * T_{\lambda,c}^{up}$ represents the TOA contribution of light reflected by a Lambertian surface with spectral albedo $R_{rs,\lambda}$, $E_\lambda^{BOA}$ is the bottom-of-atmosphere (BOA) downward-directed irradiant reflectance (a normalized irradiance, analogous to the TOA reflectance), and $T_{\lambda,c}^{up}$ is the azimuthally-averaged upward transmittance from the surface to the MISR camera of interest. *Mix_Weight* represents the individual weight given to each aerosol mixture for aggregating aerosol properties into one effective mixture, and $M_{Min}$ represents the minimum cost function among all aerosol mixtures. The choice of inverse

exponential weighting was made to avoid the use of specific mixture acceptance thresholds that would arbitrarily exclude some mixtures entirely.

### 2.1.1 Research Algorithm Detailed Description

To retrieve the remote-sensing reflectance, we prescribe the wind-driven effects of a roughened ocean surface (glint and whitecaps), using cross-calibrated multi-platform (CCMP; 0.25°, 6 hourly) 10-meter wind speed version 2 data (Atlas et al., 2011; Wentz et al., 2015). The Rayleigh scattering contribution to TOA reflectance is prescribed with a 1013.25 mb surface pressure. This allows us to remove one dimension from our look-up-table (LUT) and has minimal impact over ocean, including coastal regions (though it might have an impact over elevated inland lakes). Once the LUT has been interpolated to the appropriate solar/viewing geometry and wind speed, we then iterate through our grid of AOD, calculating the $R_{rs,\lambda}$ needed to compute $M$. This is done by taking the derivative of (2) with respect to $R_{rs,\lambda}$, setting it equal to zero and solving for $R_{rs,\lambda}$.

$$R_{rs,\lambda} = \frac{\Sigma_c\left[\frac{w_{\lambda,c}}{Unc_{\lambda,c}^2}*T_{\lambda,c}^{up}*\left(\rho_{\lambda,c}^{TOA}-\rho_{\lambda,c}^{Model}\right)\right]}{E_\lambda^{BOA}*\Sigma_c\left[\frac{w_{\lambda,c}}{Unc_{\lambda,c}^2}*\left(T_{\lambda,c}^{up}\right)^2\right]} \tag{3}$$

To prevent the algorithm from retrieving an unphysically dark surface, we establish the following $R_{rs}$ minima for the blue, green, red and NIR: [0.005, 0.003, 0.0005, 0.00008]. As there can be only one set of $R_{rs,\lambda}$ that minimizes $M$ for any given AOD and aerosol mixture, $M$ becomes inherently one-dimensional (for a given aerosol mixture). We then calculate $M$ for every AOD on the AOD LUT grid until $M$ begins to increase, which is then used to calculate the first ($M'$) and second ($M''$) derivatives of $M$ with respect to AOD at the minimum value of $M$ for each mixture. The AOD reported by each aerosol mixture is obtained from using Newton's method, defined as

$$\tau_{new} = \tau_{old} - \frac{M'}{M''}. \tag{4}$$

$\tau_{new}$ is the mixture-reported AOD and $\tau_{old}$ is the AOD corresponding to the minimum value of $M$. Once $M$ is minimized for each aerosol mixture, we then aggregate acceptable aerosol mixture results by weighting AOD, aerosol properties, and surface properties according to 2b, so better-fitting mixture/AOD/surface combinations are weighted more heavily.

### 2.1.2 RA Glint Screening

Building on the glint screening improvements made in *Limbacher and Kahn* [2017], we use a two-tier approach to account for sun-glint. The first tier is a simple glitter angle test:

$$glitter_c = min\left\{max\left[\frac{(G_c-10.0)}{20.0-10.0},0.0\right],1.0\right\}, \tag{5}$$

where $G_c$ represents the glitter angle relative to the view angle for a particular camera $c$. This equation yields a value of 0 when the glitter angle is within 10˚ of the view angle, a value of 1 when the glitter angle exceeds 20˚ of the view angle, and values spread linearly between 0 and 1 for intermediate glitter angles. Once computed, this weight is then multiplied into $w_{\lambda,c}$. As the algorithm now uses information from all cameras with glitter angles as low as 10 degrees, we must include the

effects of sun-glint in our estimate of uncertainty. To speed up computation, the estimate of sun-glint uncertainty for a given band and camera is computed for an aerosol-free atmosphere as the following:

$$Unc_{\lambda,c}^{Gl} = \sqrt{max\{|\rho_{\mu,\mu_0,d\Phi,ws}^{Model} - \rho_{\mu+\Delta\mu,\mu_0+\Delta\mu_0,d\Phi+\Delta d\Phi,ws+\Delta ws}^{Model}|\}^2 + \left(\frac{\rho_{\mu,\mu_0,d\Phi,ws}^{Glint}}{10}\right)^2}. \tag{6}$$

Here $\rho_{\mu,\mu_0,d\Phi,ws}^{Model}$ represents the modeled reflectance for any given band and camera, interpolated to the observed solar/viewing geometry and wind-speed for a particular location in the MISR swath. $\mu$ represents the cosine of the viewing-

zenith angle, $\mu_0$ represents the cosine of the solar-zenith angle, $d\Phi$ represents the relative-azimuth angle between the solar and viewing vectors, and $ws$ represents the CCMP interpolated wind speed. $\rho_{\mu+\Delta\mu,\mu_0+\Delta\mu_0,d\Phi+\Delta d\Phi,ws+\Delta ws}^{Model}$ represents the model reflectance for a perturbed solar/viewing geometry and perturbed wind-speed. The perturbation is 3 m/s for wind speed, 0.01 for $\mu_0$ and $\mu$, and 2° for $d\Phi$; this term represents an attempt to account for uncertainty in glint modeling due to wind-speed error, anisotropy, etc. $\rho_{\mu,\mu_0,d\Phi,ws}^{Glint}$ represents the difference between the modeled TOA reflectance over a Fresnel

reflector with whitecaps (dependent on $ws$), and the modeled TOA reflectance over a black surface. This last term in equation 6 is just 10% of the modeled surface-attenuated reflectance in the band and camera (squared), and is our guess at a lower bound on the reflectance uncertainty associated with sun glint.

### 2.1.3 RA Input Uncertainties

Previous versions of the MISR RA (as well as the SA) have attempted to characterize uncertainty only in terms of TOA

observed reflectance, rather than attempting to account for uncertainty due to stray-light or forward modeling errors (such as glint). Here, we employ a heuristic approach by attempting to account for the relative magnitude of some of the largest uncertainties (the actual coefficients used for the uncertainties are educated guesses), though we neglect the fact that these uncertainties are at least partly correlated.

$$Unc_{\lambda,c} = \sqrt{\left(Unc_{\lambda,c}^{TOA}\right)^2 + \left(Unc_{\lambda,c}^{Gl}\right)^2 + \left(Unc_{\lambda,c}^{SL}\right)^2} \tag{7}$$

$$Unc_{\lambda,c}^{TOA} = \sqrt{\left(0.04 * \rho_{\lambda,c}^{TOA}\right)^2 + (0.002)^2}$$

$$Unc_{\lambda,c}^{SL} = \sqrt{\left(f_c * 0.01 * [\rho_{\lambda,c}^{TOA} - \rho_{\lambda,c}^{BG}]\right)^2}$$

$Unc_{\lambda,c}^{TOA}$ represents the uncertainty in TOA reflectance measurement, *estimated* here as a spectrally invariant 4 percent relative and 0.002 absolute. $Unc_{\lambda,c}^{Gl}$ represents our estimate of glint uncertainty, described in 2.1.2. As $\rho_{\lambda,c}^{BG}$ represents the

swath-averaged value of $\rho_{\lambda,c}^{TOA}$, $Unc_{\lambda,c}^{SL}$ represents the uncertainty in our stray-light correction, represented here as uniform "veiling-light" with the uncertainty set to one percent magnitude for the nadir camera. Work by *Witek et al.* [2017] indicates that the veiling-light error for MISR seems to increase with view angle, so we choose $f_c$ to be [6, 2.5, 1.5, 1, 1, 1, 1.5, 2.5, 6] going from the 70˚ forward through nadir to the 70˚ aft-viewing cameras with the additional veiling-light term set to 0.01 at

nadir. As the additional veiling-light correction we apply here is identical to our stray-light uncertainty, we are indicating that there is a high degree of uncertainty associated with this correction (and our stray-light correction in general). Our channel-specific weights, $w_{\lambda,c}$ are initially set to unity, but are then multiplied by the result of equation 5 above.

## 2.2 The AErosol RObotic Network (AERONET)

AERONET sun photometers (*Holben et al.*, 1998) provide direct measurements of spectral AOD with nominal uncertainty of

±0.01 (*Eck et al.*, 1999; *Sinyuk et al.*, 2012), verified by periodic reference calibration. Ångström exponent (ANG), which represents the slope of a line fitted to the log of AOD vs. the log of wavelength, is also accurately reported as long as the spectral AOD is sufficiently large (*Wagner and Silva*, 2008). AERONET almucantar inversions (*Dubovik and King*, 2000) provide constraints on coarse-mode sphericity (*Dubovik et al.*, 2006) and column-effective aerosol single scattering albedo (SSA), provided the aerosol loading is high (AOD at 440 nm > 0.4), the sun is low in the sky, the aerosol is spatially uniform

(*Holben et al.*, 2006), and the surrounding surface is not highly reflective (*Sinyuk et al.*, 2007). Interpretation of the almucantar inversions is complicated when multiple aerosol modes reside in the column [*Schuster et al.*, 2016].

For comparison with the MISR RA (section 3 below), we first interpolate AERONET AOD to the MISR band centers, using a second-order polynomial in log-space: 446 nm (blue), 558 nm (green), 672 nm (red), and 866 nm (NIR). ANG is then

computed as a log-log fit of interpolated AOD to wavelength (using all 4 wavelengths). To limit AOD spatial and temporal variability from impacting comparisons with MISR, we limit potential coincidences to ±30 minutes from the MISR overpass time, saving mean spectral AOD, computing ANG from these values, and retaining the AOD deviation (max-min over time period) for all four MISR spectral bands.

## 2.3 The MODIS-Terra Ocean-Color (OC) Product

Because a main contribution of this paper is the validation of AOD and Ångström Exponent from the MISR RA over complex water scenes with the MISR RA, we also compare results from the MISR RA to those from the MODIS-Terra ocean-color product for a few selected scenes. To do this, we first re-grid the MODIS-Terra data ($R_{rs}$, AOD, ANG, and level-2 mask data [https://oceancolor.gsfc.nasa.gov/atbd/ocl2flags/]) to the MISR 1.1km L1B2 projection via a nearest-neighbor approach. We then mask all data that is flagged in the MODIS level-2 product (there tend to be far fewer retrievals

meeting level-3 criteria) or falls outside of the MISR nadir-camera swath. $AOD_{869}$ (NIR) is converted to $AOD_{558}$ using the Ångström Exponent provided in the MODIS product. Details of the algorithms used to generate MODIS OC $R_{rs}$ (AOD and

ANG are retrieved as part of this process) are briefly summarized in section 1 above, and are described in detail at https://oceancolor.gsfc.nasa.gov/atbd/rrs (last accessed 1/16/2019).

## 2.4 The MISR V23 Standard Aerosol Product

In addition to comparing the MISR RA to the MODIS OC product, we also compare the RA to the recently released MISR

version 23 standard aerosol product (SA).  Recent upgrades to the SA have resulted in improved spatial resolution (from 17.6km to 4.4 km; *Garay et al.*, 2017), better AOD statistics over ocean due to a stray-light correction (*Witek et al.*, 2017), and the inclusion of pixel-level AOD retrieval uncertainty (over dark water only; *Witek et al.*, 2018).  Because the v23 SA reports AOD at 550 nm, we first calculate the SA AOD for all four spectral bands using the scaling coefficients provided in the v23 product, and then calculate the Ångström Exponent using all four MISR wavelengths (as described in 2.2 above).

No additional cloud screening is required for the MISR SA (provided one does not use the 'raw' data product, which provides AOD with minimal cloud screening).

## 3. Example MISR Aerosol-Surface Retrievals Over Ocean

Although much information can be gleaned from the statistical validation of MISR-retrieved spectral AOD against AERONET, scene analysis provides some useful context.  In Figure 1 we present MISR RA, MODIS Terra OC,

and MISR SA results for the Florida Strait region on December 22, 2012, including retrievals over bright (and very shallow) water.  Figure 1a demonstrates the capability of the MISR RA to retrieve AOD even over the very shallow waters off the western coast of Florida.  (The MISR RA $R_{rs}$ 558 nm / 446 nm ratio (panel 1c) highlights the difference in water spectral reflectance off the Florida west coast vs. elsewhere in the region.)  Compared to the MISR RA, the MISR V23 SA, and AERONET, MODIS OC AOD (panel 1e) appears to be biased very high

(>100% compared to MISR) in the eastern half of the scene, likely due to inaccurate forward modeling of sun-glint or whitecaps.  Also, the MODIS OC ANG values (panel 1f) appear to be skewed low compared to MISR and AERONET.  MISR V23 SA AOD and ANG agree well with the RA in the few portions of the scene where the SA reports aerosol information, but the V23 SA masks aerosol retrievals in locations with shallow or turbid water, resulting in few retrievals over the region. Panels 1c and 1g show how the differences in retrieved aerosol properties

(between the RA and MODIS) can propagate into the retrieval of $R_{rs}$, and how the MISR RA atmospheric characterization allows increased coverage over near-coastal water. The retrieved $R_{rs}$ for the blue band (not shown) approaches 0.3 for the region just north of Key West, which gives an indication of the versatility of the RA algorithm to retrieve over shallow water when the ocean surface is quite bright.

Figure 2 shows results for the northern Bay of Bengal on January 29th, 2015, including results over notoriously turbid water.  The RA indicates that the entire region is dominated by small-medium (ANG ~ 1.0-1.2) spherical non-absorbing aerosol particles (shape and light-absorption retrieval plots not shown), consistent with expected pollution particles in this region.  Even with 558 nm AOD approaching 0.5, the algorithm is able to separate the surface from the atmosphere, as shown across the first row of Figure 2.  Qualitatively, the MODIS OC and MISR RA AOD and $R_{rs}$ agree well with each other, although MODIS suggests that the aerosol might be a bit smaller than the RA reports (panel 2b compared to 2f), and that the water might be a little less green (2c compared to 2g). Although the MISR RA AOD in panel 2a does not appear to be correlated with the water color seen in panel 2d, indicating successful surface-atmosphere separation, it does appear as though retrieved ANG might be biased a bit low in the turbid water regions (discussed further in section 4 below).  AOD and ANG retrieved from the MISR RA and MISR SA agree extremely well with each other, even though the comparisons are confined to the non-turbid water regions of the scene where the SA provides coverage. Again, this figure illustrates and how the MISR RA atmospheric correction allows for increased $R_{rs}$ coverage over near-coastal water.

Figure 3 shows results for the eutrophic northern Caspian Sea on September 19, 2015.  The RA (panel 3b) indicates that a plume comprised of small (ANG ~1.5), spherical non-absorbing (shape and light-absorption retrieval plots not shown) aerosol is present over a ~20,000 km$^2$ region in the north-central part of the plotted domain.  Within the plume, the RA indicates that the mid-visible AOD varies between 0.10 and 0.15, whereas outside the plume the retrieved AOD is ~ 0.05.  This skill in identifying relatively low AOD plumes over bright water is due to the multiple view angles provided by MISR.  Because the portion of TOA reflectance attributed to the retrieved $R_{rs}$ decreases with increasing view angle, the spatial pattern of the MISR Df (70˚ forward view angle) NIR reflectance (panel 3k) correlates well with the retrieved AOD (panel 3a).  Although the MISR RA shows skill separating the effects of atmospheric and oceanic scattering, Figures 3b and 3d suggest some surface artifacts might be aliasing the ANG retrieval.  That said, the artifacts still do not lead to substantially different AOD retrievals between the dark-water and eutrophic-water portions of the scene.  MODIS actually masks much of the eutrophic region for this scene, with large discrepancies found between the MISR RA and MODIS for much of the remaining portions.  The MISR SA does not report any aerosol retrievals over water in this region, probably due to shallow water masking.

In Figures 4 and 5 we present results from the MISR RA, MODIS OC algorithm, and the MISR SA for the eastern Yellow Sea region on March 13, 2012.  Note that although AERONET data are plotted for this region, results over land may not be representative of the air column over nearby ocean, particularly if the AERONET station is elevated.  Panels 4a and 4e show good AOD agreement between the MISR RA and MODIS, although panels 4b and 4f indicate rather large discrepancies in ANG between the two retrievals.

Although AERONET data from the DRAGON deployment in S.E. Asia show variability in ANG (~1.1-1.5), MISR ANG appears low-biased and MODIS ANG appears slightly high-biased. Not surprisingly, this discrepancy between MISR and MODIS ANG shows up in the $R_{rs}$ ratios (panels 4c and 4g, respectively). Note that in the southwest corner the of the scene, where the MISR RA retrieves AOD of up to ~1, the MISR RA is still capable of retrieving water color (panels 4d and 4l), whereas MODIS does not provide results for this part of the scene. In Figure 5, we present the MISR-retrieved $R_{rs}$ RGB for this scene juxtaposed with a Rayleigh-corrected MISR nadir true-color image and the MODIS retrieved $R_{rs}$ RGB.

Additional examples are included in Supplemental Material: a biological bloom in the East Argentine Sea, and three cases over the Bohai Sea along coastal China east of Beijing, showing results for low, moderate, and high AOD loading.

## 4 Statistical Validation of the MISR Over-Water RA AOD and ANG Using AERONET

As in *Limbacher and Kahn* [2017], MISR RA aerosol retrievals are performed at 1.1 km resolution. The newly released MISR V23 aerosol product now reports AOD and aerosol properties at 4.4 km resolution, a major improvement compared with the 17.6 km resolution for the version 22 (*Garay et al.,* 2017). For validation against AERONET, we average all good-quality MISR 1.1 km retrievals surrounding the AERONET site, yielding one value per coincidence. We consider AERONET/MISR coincidences only if the following conditions are met:

- At least one AERONET observation on each side of the temporal averaging window (±30 minutes of MISR overpass).
- AERONET temporal variability (max-min for all 4 MISR bands) < 0.05 + 0.1*AOD.
- AERONET reported elevation <100 meters. This is necessary for two reasons:
    - AERONET AOD (and ANG) might otherwise not be comparable to our over-water retrievals (ocean retrievals are all at sea-level).
    - The LUT we use (for water retrievals only) contains only one surface pressure value (1013.25 mb), meaning that retrievals over elevated inland lakes could suffer from non-negligible errors, depending on elevation.
- At least 5% of MISR observations within 25 km of the AERONET site result in good-quality MISR over-water aerosol retrievals. A good-quality MISR aerosol retrieval requires **all** the following pixel-level criteria to be met (most of which are for cloud screening):
    - MISR RA cost function ($M$) < 1, which indicates a good model fit to the observations.
    - MISR RA maximum channel-specific cost function < 0.5, to screen out clouds that might only be visible in one or two cameras.

o   MISR RA $M/M'' < 10^{-3}$, as this ratio tends to increase when clouds are present.

o   Additionally, to improve cloud detection, we flag all MISR retrievals immediately surrounding any pixel whose aerosol retrieval does not meet the three quality thresholds outlined above.

These constraints yield 2,419 MISR-AERONET coincidences for the four years of MISR data we currently have processed
(four years interspersed between September 2000 and November 2016).

Results from the statistical comparison of MISR and AERONET are shown in Tables 1 and 2 (558 nm AOD and ANG, respectively). In order to identify co-variation between retrieved surface albedo and aerosol properties, we perform the following comparisons for both AOD and ANG against AERONET:

- *Average* – Average all good-quality MISR aerosol retrievals within 25 km of the AERONET site,
- *Lowest 10%* – Average only those good-quality MISR aerosol retrievals where the retrieved 558 nm (green) $R_{rs}$ is lower than the $10^{th}$ percentile value for that specific scene,
- *Highest 10%* – Average only those good-quality MISR aerosol retrievals where the retrieved 558 nm (green) $R_{rs}$ is higher than the $90^{th}$ percentile value for that specific scene.

Figure 6 illustrates how the data are parsed and averaged; it provides some context for the AOD and ANG scatterplots presented in Figure 7, which shows both scene-averaged values for ANG and AOD in addition to 2-d histograms of AOD and ANG errors as a function of water color. We create the following productivity/turbidity index (PTI) that allows us to characterize MISR retrieval errors against a single water-color parameter (similar to NDVI over land):

$$PTI = \frac{MISR\ [Green+Red+NIR]\ R_{rs} - MISR\ Blue\ R_{rs}}{MISR\ R_{rs}\ Spectral\ Sum}. \qquad (8)$$

Note that PTI is not retrieved, rather it is calculated from the retrieved spectral remote-sensing reflectances. Figure 6b combined with 6d illustrate a limitation of our cloud screening, as we clearly fail to screen all the clouds in the upper-left portion of the plots. Additionally, the large AOD variability found within the 25 km averaging window suggests that some of the difference between AERONET and MISR (Figure 7 and Tables 1 and 2) is likely related to scene spatial variability,
combined with the differences in MISR and AERONET spatial-temporal sampling, a conclusion reached in earlier research as well (*Kahn et al.*, 2010).

Figure 7(a, c) and Table 1 demonstrate the ability of the MISR RA to retrieve AOD under a variety of conditions: low AOD, high AOD, over oligotrophic, eutrophic, shallow, and turbid water. Specifically, Figure 7c illustrates the general
insensitivity of retrieved AOD to water color. This is further demonstrated by the eight color patches at the top of Figure 7 panels (c) and (d). They present the averaged MISR RA-retrieved $R_{rs}$ values (made into RGB images and enhanced 10x in brightness) for the corresponding PTI range along the x axis of the panel. For PTI ~ -1, the water is oligotrophic, and its

color is blue. As PTI increases to values between about 0 and 0.5, the water tends toward eutrophic and begins to appear green. As PTI exceeds about 0.75, the water appears more brown than green, and would likely be classified as turbid.

Table 1 appears to confirm the insensitivity of retrieved AOD to water color, as the AOD statistics from the darkest waters of the scene are nearly identical to those over the brightest waters. We expect brighter, more productive waters near the coast, where AERONET stations are preferentially located. The average "bright water" retrieval was performed an average of 17.3 km from the AERONET location; the dark(er) water retrievals were performed 19.8 km from AERONET on average, i.e., generally including darker water farther from the coast. So the darker water retrievals are more likely to encompass a gradient in aerosol amount. This might explain why Table 1 seems to indicate that at low AOD, the brighter albedo retrievals tend to perform better than the lower $R_{rs}$ retrievals. Overall, 72% of aerosol retrievals fall within the maximum of 0.03 or 10% of the AERONET AOD value. Figure 7(a, c) and Table 1 indicate that the MISR RA is quite capable of retrieving good-quality AODs over scenes that would challenge many other aerosol retrieval algorithms.

In Figure 7(b, d) and Table 2, we compare ANG results from the MISR RA to those calculated from AERONET AODs, for scenes where the AERONET 558 nm AOD > 0.20. Because ANG is more meaningful as AOD increases, it is more difficult to assess the sensitivity of RA-retrieved ANG to RA-retrieved surface albedo, as there are only have 502 coincidences for AOD > 0.20 in the validation dataset. Nevertheless, it does appear as though the uncertainty of RA-retrieved ANG increases as the water becomes greener (or more turbid). This can be seen qualitatively in Figure 7d based on increased scatter density with increasing PTI, and quantitatively by comparing the bright with the dark water retrievals in Table 2. Some of this might be due to the limited range of light-absorbing particle properties in the current algorithm climatology (*Kahn et al*., 2010; *Limbacher and Kahn*, 2014). Still, with a linear correlation coefficient of 0.89 and an RMSE of 0.25, the current MISR RA provides some quantitative information about particle size.

## 5. Conclusions

In *Limbacher and Kahn* [2017], we demonstrated results from an upgraded MISR Research Aerosol retrieval algorithm (RA) that allows us derive *Chlorophyll-a* concentration from MISR over Case I waters. We also presented modifications to a stray-light correction algorithm first detailed in *Limbacher and Kahn* [2015] and identified (and corrected) temporal trends in the MISR radiometric calibration. Here, after applying those corrections to the MISR top-of-atmosphere radiances, we upgrade and extend the MISR RA for use over turbid, shallow, and eutrophic waters. By assuming the water-body reflectance can be modeled adequately as a Lambertian reflector, this upgraded 1.1km resolution retrieval algorithm self-consistently retrieves AOD, aerosol type, and ocean surface remote-sensing reflectance ($R_{rs}$).

We introduce a productivity/turbidity index (PTI), calculated from retrieved spectral $R_{rs}$, that distinguishes water types (similar to NDVI over land), and validate the upgraded RA statistically by comparing MISR retrieved spectral AOD and Ångström exponent (ANG) against 2,419 cloud-screened coincident direct-sun observations from AERONET. We also compare MISR RA results with those from the MODIS ocean color algorithm (OC) and the MISR Standard Aerosol retrieval

algorithm (SA) for several turbid, shallow, and eutrophic water cases. Results indicate that the MISR RA has sensitivity to both AOD and ANG, even over bright waters, and typically out-performs the MODIS OC and MISR SA in coverage, or in accuracy compared to expectation. For mid-visible AOD retrieved under all conditions, we report a root-mean-squared-error (RMSE) of 0.038 and a linear-correlation coefficient (r) of 0.95. For the 502 scenes where AERONET AOD > 0.20, we report an ANG RMSE of 0.25 and r=0.89. Although AOD results do not appear to be substantially affected by water color,

uncertainty associated with our ANG retrievals appear to increase as the water "greens" (or "browns").

The "atmospheric correction" needed to obtain accurate surface reflectance retrievals should be much more robust from the MISR multi-angle instrument than from single-view instruments such as MODIS. As such, MISR data could be used to make substantial contributions to the ocean color community. This is especially true in regions of shallow, turbid, and

eutrophic waters, locations where aerosol retrieval algorithms for single-view instruments tend to be under-determined, yet waters tend to be more biologically productive than the oligotrophic waters where single-view retrievals perform best. Therefore, a next step forward from this work would be to use the MISR RA to constrain the atmospheric-correction algorithm used by the ocean-color community for OC retrievals, possibly extrapolated in space and time when applied to single-view instruments that offer greater coverage and include spectral bands optimized for ocean color retrievals. We hope

this will result in more robust and numerous OC retrieval results over biologically productive coastal waters and major aerosol transport pathways over ocean.

**Conflict of interest**

The authors declare that they have no conflict of interest.

**Acknowledgments**

We thank Chris Proctor and NASA's Ocean Biology Processing Group for providing the MODIS–Terra ocean color products. We thank our colleagues on the Jet Propulsion Laboratory's MISR instrument team and at the NASA Langley Research Center's Atmospheric Sciences Data Center for their roles in producing the MISR Standard data sets, and Brent Holben at NASA Goddard and the AERONET team for producing and maintaining this critical validation dataset. CCMP Version-2.0 vector wind analyses are produced by Remote Sensing Systems. Data are available at www.remss.com. This

research is supported in part by NASA's Climate and Radiation Research and Analysis Program under Hal Maring, and NASA's Atmospheric Composition Program under Richard Eckman.

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

**Table 1: Statistics of MISR 558 nm (Green) AOD retrievals as compared to AERONET.**

| 0.00<AERONET AOD<1.00 | r | MAE | RMSE | Bias | 0.03 or 10% | # |
|---|---|---|---|---|---|---|
| Average (<25 km) | 0.954 | 0.018 | 0.038 | 0.006 | 0.717 | 2419 |
| Lowest 10% $R_{rs}$ (<25 km) | 0.938 | 0.021 | 0.044 | 0.010 | 0.649 | 2419 |
| Highest 10% $R_{rs}$ (<25 km) | 0.949 | 0.020 | 0.041 | 0.010 | 0.678 | 2419 |
| | | | | | | |
| 0.00<AERONET AOD<0.10 | r | MAE | RMSE | Bias | 0.03 or 10% | # |
| Average (<25 km) | 0.654 | 0.015 | 0.024 | 0.010 | 0.818 | 1202 |
| Lowest 10% $R_{rs}$ (<25 km) | 0.573 | 0.017 | 0.032 | 0.015 | 0.735 | 1202 |
| Highest 10% $R_{rs}$ (<25 km) | 0.641 | 0.016 | 0.027 | 0.012 | 0.780 | 1202 |

In this table, r is the Pearson correlation coefficient, MAE is the median absolute error, RMSE is the root mean squared error between the satellite retrieval and AERONET, bias is the mean MISR-AERONET value, 0.03 or 10% represents the fraction of MISR retrievals falling within the maximum of 0.03 or 10% of the AERONET AOD and # is the number of validation cases included. In the first column, average refers to using the average MISR retrieval value (good-quality) over all retrievals within 25 km of the AERONET site, the lowest 10% $R_{rs}$ refers to using only the pixels with $R_{rs}$ (green band only) lower than the 10th percentile values (for a given cloud-screened scene) for comparison with AERONET, and the highest 10% $R_{rs}$ refers to using only the retrievals with the 10% largest $R_{rs}$ for comparison with AERONET.

**Table 2: Statistics of MISR ANG retrievals as compared to AERONET.**

| 0.00<AERONET AOD<1.00 | r | MAE | RMSE | Bias | # |
|---|---|---|---|---|---|
| Average (<25 km) | 0.662 | 0.250 | 0.416 | -0.133 | 2419 |
| Lowest 10% $R_{rs}$ (<25 km) | 0.643 | 0.257 | 0.416 | -0.112 | 2419 |
| Highest 10% $R_{rs}$ (<25 km) | 0.640 | 0.274 | 0.439 | -0.180 | 2419 |
| | | | | | |
| 0.20<AERONET AOD<1.00 | r | MAE | RMSE | Bias | # |
| Average (<25 km) | 0.890 | 0.169 | 0.250 | -0.025 | 502 |
| Lowest 10% $R_{rs}$ (<25 km) | 0.885 | 0.173 | 0.252 | -0.003 | 502 |
| Highest 10% $R_{rs}$ (<25 km) | 0.861 | 0.194 | 0.282 | -0.060 | 502 |

**In this table, the statistical parameters (e.g. RMSE, etc) are defined the same way as in Table 1.**

FIGURE CAPTIONS

**Figure 1.** Example of MISR RA aerosol retrieval over *shallow water*. MISR imagery acquired on December 22, 2012, 16:07Z: Terra Orbit 69220, Blocks 70-71, over southern Florida and the Florida Strait. MISR RA 558 (Green) nm AOD, MODIS Terra Ocean Color (OC) AOD (558 nm, extrapolated using MODIS ANG), and the MISR version 23 SA AOD (558 nm) are presented in the first column (panels a, e, i, respectively). Plots of MISR RA ANG, MODIS Terra OC ANG, and MISR version 23 SA ANG are provided in the second column (panels b, f, j). MISR RA $R_{rs}$ ratios (558 nm / 446 nm), MODIS OC $R_{rs}$ ratios (555 nm / 443 nm), and the MISR 70.5° forward viewing near-infrared (NIR) reflectance are provided in the third column (panels c, g, k, respectively). Brightness enhanced true-color images of MISR RA $R_{rs}$, MODIS Terra OC $R_{rs}$, and a MISR nadir context image (Rayleigh corrected), are presented in the 4th column (panels d, h, l, respectively). Where available, AERONET data are embedded in the plots as shaded circles matched to the scale of the relevant color bar.

**Figure 2.** Same as Figure 1, except for *turbid water* found in the Northern Bay of Bengal. MISR imagery acquired on January 29, 2015, 04:36Z: Terra Orbit 80397, Blocks 73-75.

**Figure 3.** Same as Figure 1, except for *eutrophic waters* found in the northern Caspian Sea. MISR imagery acquired on September 19, 2015, 07:42Z: Terra Orbit 83792, Blocks 53-55. Note that there are no MISR SA V23 over-water retrievals for this region.

**Figure** 4 Same as Figure 1, except for the ***optically diverse waters*** of the Yellow Sea. MISR imagery acquired on March 13, 2012, 02:30Z: Terra Orbit 65076, Blocks 60-64. Coincident AERONET data were acquired as part of the Southeast-Asian leg of the DRAGON field campaign (colored circles). As most AERONET data are taken over land (some elevated), the data are not necessarily representative of the adjacent oceanic air column.

**Figure 5**. MISR (and MODIS) retrieval imagery acquired on March 13, 2012, 02:30Z: Terra Orbit 65076, Blocks 60-64. RGB images have been brightness-enhanced. The center plot represents the MISR An True-color Rayleigh-corrected RGB for the scene. The left plot represents the MISR RA retrieved $R_{rs}$ RGB image, and the right plot represents the MODIS retrieved $R_{rs}$ RGB image.

**Figure 6**. Example set of 80x80 pixel MISR aerosol retrievals centered over the Dunkerque AERONET site on 1/29/2015, showing the effects of scene variability on the MISR-AERONET comparison. Brightness enhanced True-color images for the (a) nadir and (b) 70.5° forward camera. (c) Brightness enhanced true-color image of MISR RA-retrieved $R_{rs}$ and (d) MISR RA-retrieved over-water AOD. The red circle in panel (d) represents the 25 km averaging window used for MISR/AERONET comparisons. The ±30-minute average AERONET 558 nm (interpolated) AOD is provided for context. Cloud/land masking has been applied to panels c and d.

**Figure 7.** MISR vs. AERONET AOD and ANG statistical comparisons for all cloud-screened coincidences. (a) MISR vs AERONET green-band (558 nm, interpolated for AERONET) AOD data; (b) MISR vs AERONET ANG data, conditioned on AERONET 558 nm AOD >0.20. (c) 2-d histogram of all MISR-AERONET AOD differences vs. MISR retrieved PTI (equation 8). For this panel (and d), we compare all cloud-free MISR data to AERONET, meaning there can be over 1,000 MISR-AERONET data points for each unique AERONET observation. (d) 2-d histogram of all MISR-AERONET ANG differences vs. MISR retrieved PTI (equation 8). For this panel (and d), we compare all cloud-free MISR data to AERONET, meaning there can be over 1,000 MISR-AERONET data points for each unique AERONET observation. For panels c and d, the eight colored rectangular boxes represent the averaged MISR RA-retrieved $R_{rs}$ values (Brightness enhanced by 10X and made into RGB images) over the x-axis range that they physically intersect in the plots.

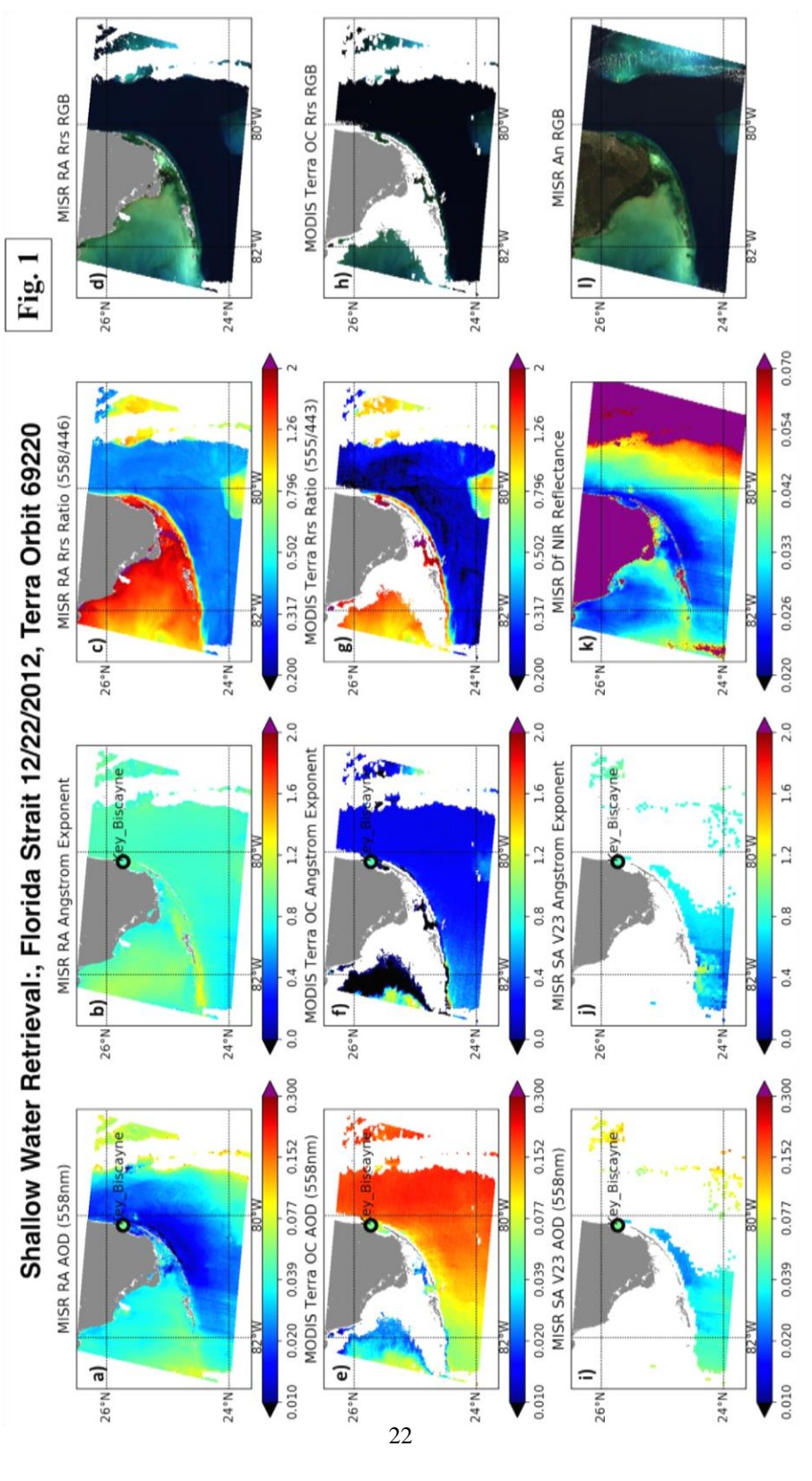

Fig. 1

Shallow Water Retrieval:, Florida Strait 12/22/2012, Terra Orbit 69220

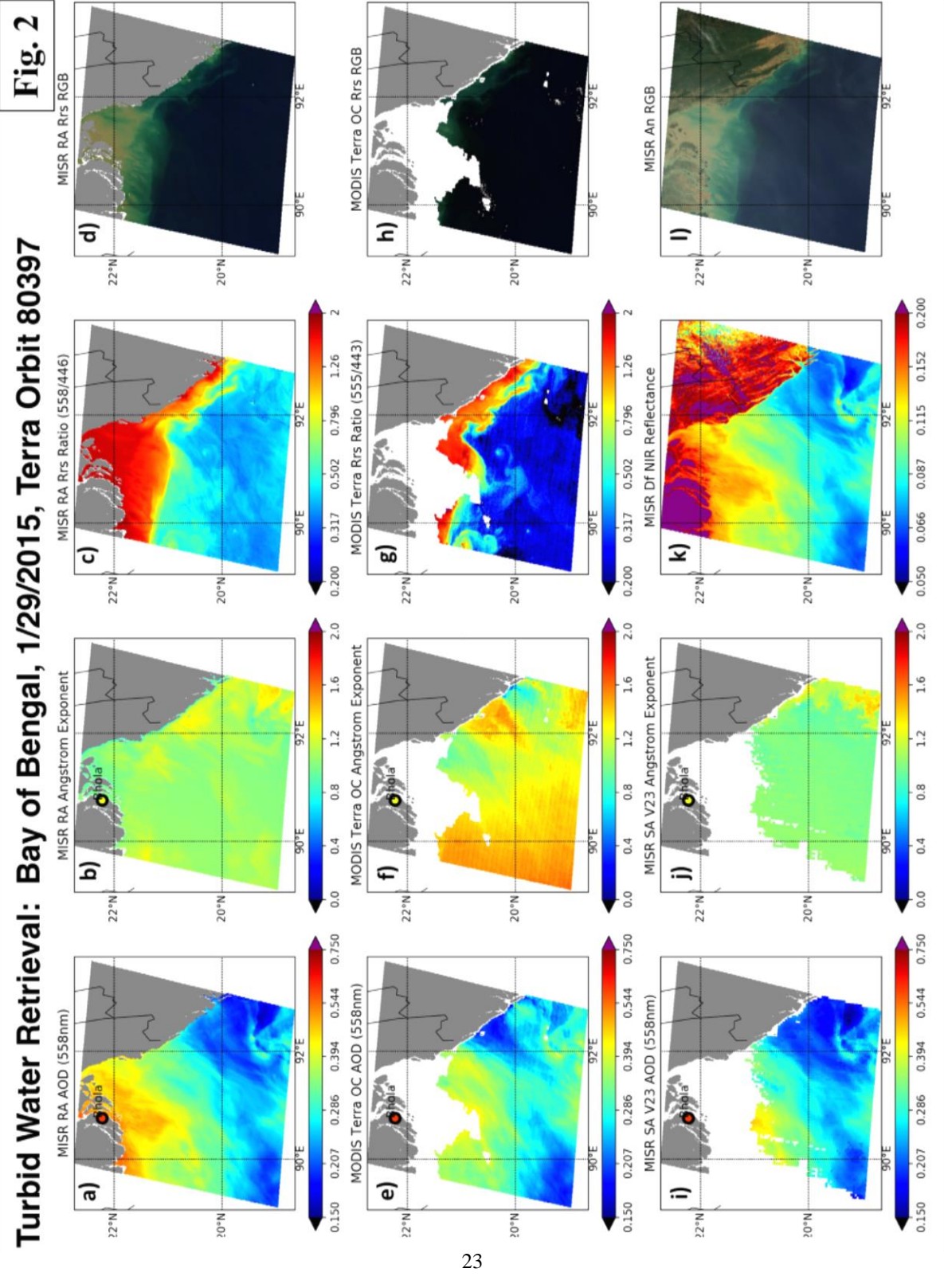

**Turbid Water Retrieval: Bay of Bengal, 1/29/2015, Terra Orbit 80397**

Fig. 2

# Eutrophic Water Retrieval:  Caspian Sea, 9/19/2015, Terra Orbit 83792

## Fig. 3

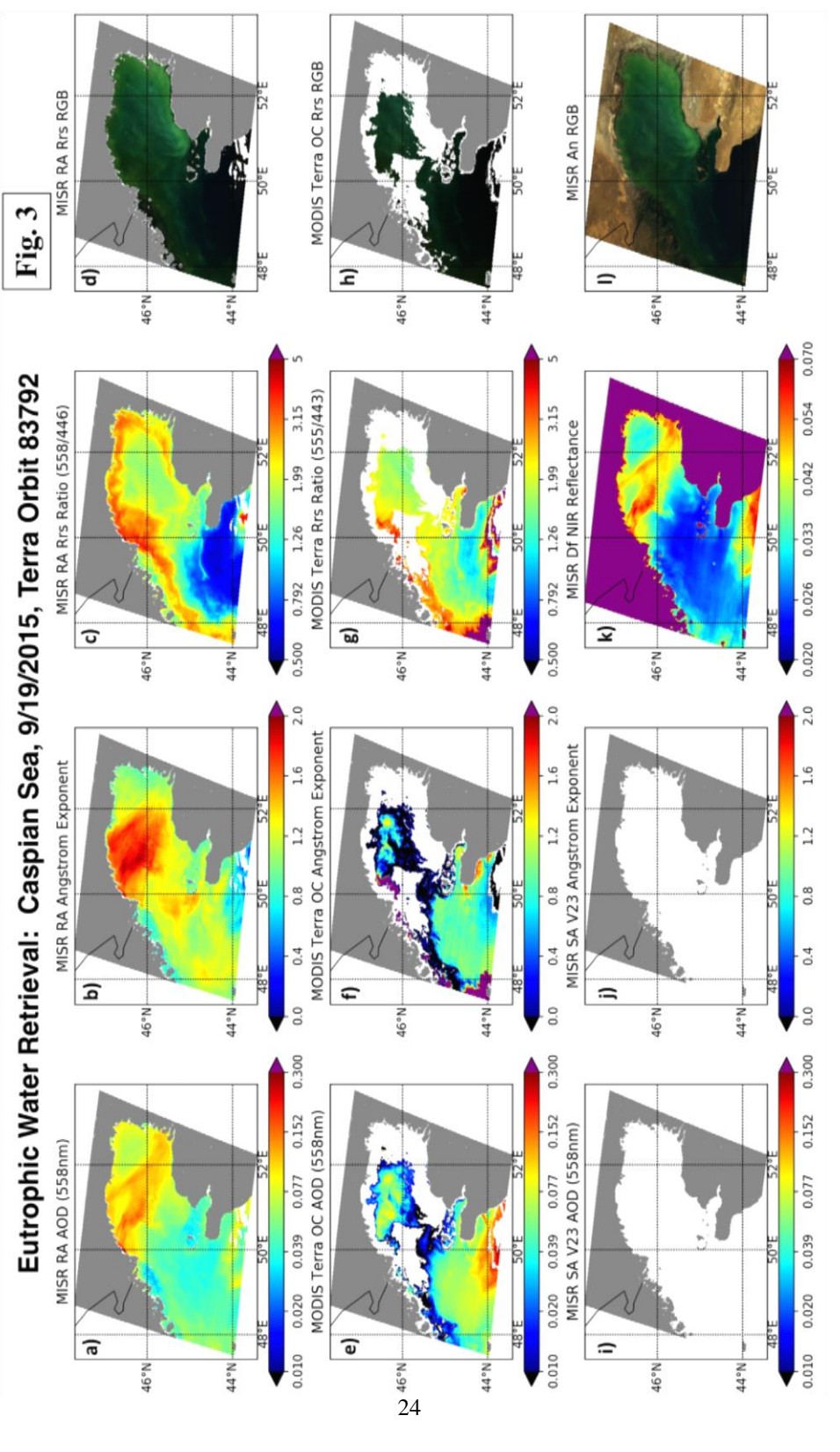

# Complex Water Retrieval: Yellow Sea, 3/13/2012, Terra Orbit 65076

## Fig. 4

# Complex Water Retrieval:  Yellow Sea, 3/13/2012, Terra Orbit 65076

Fig. 5

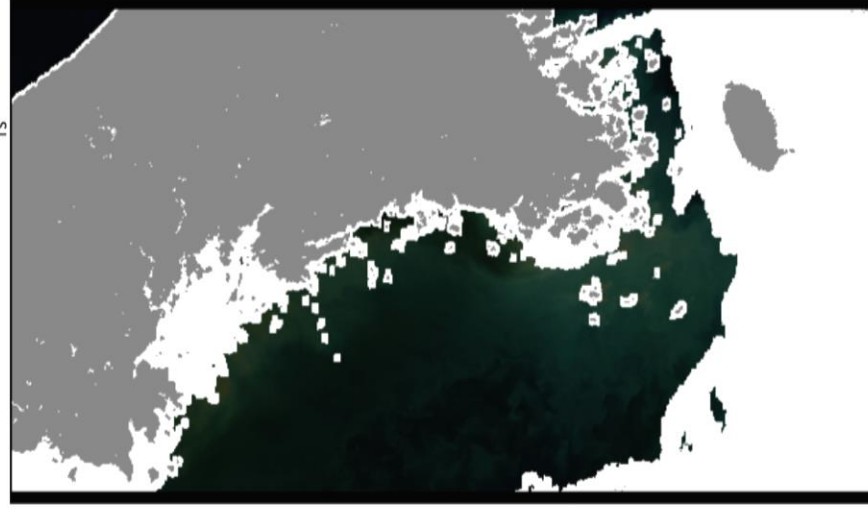

MODIS Retrieved $R_{rs}$

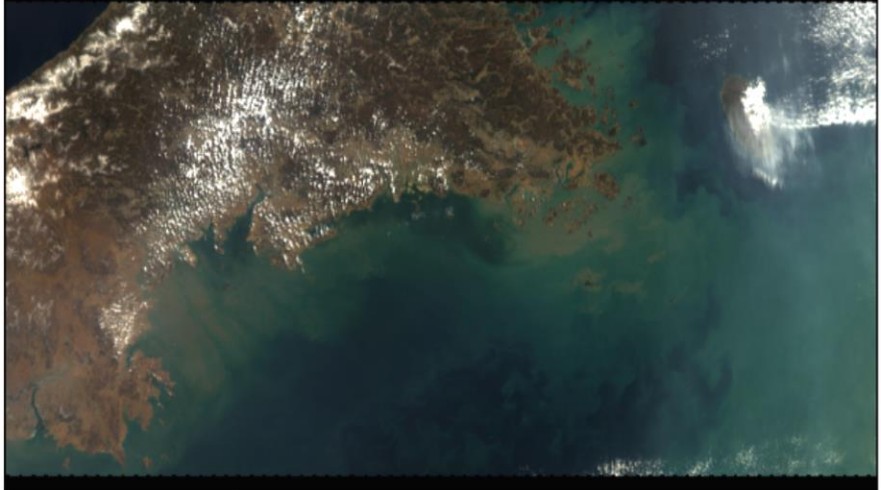

MISR An True Color Image

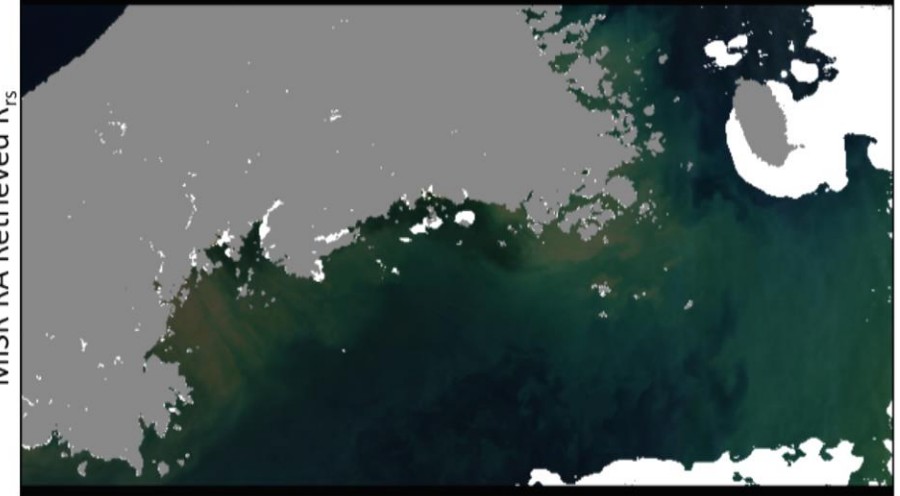

MISR RA Retrieved $R_{rs}$

# MISR/AERONET Comparison:  Dunkerque, 1/29/2015 | Fig. 6

### MISR An RGB

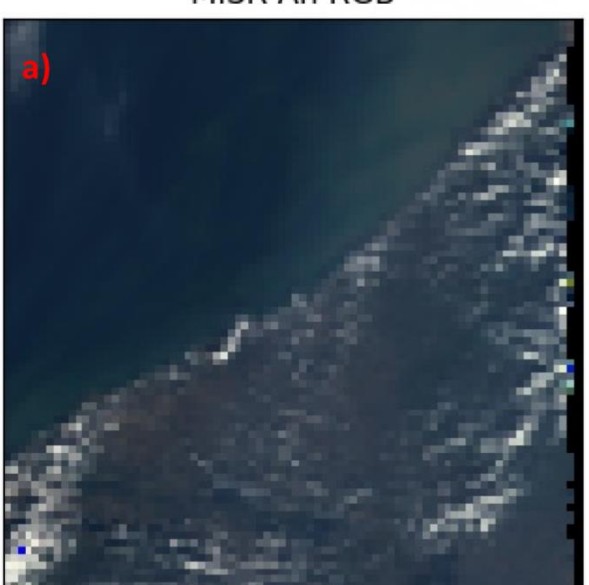

### MISR Df RGB

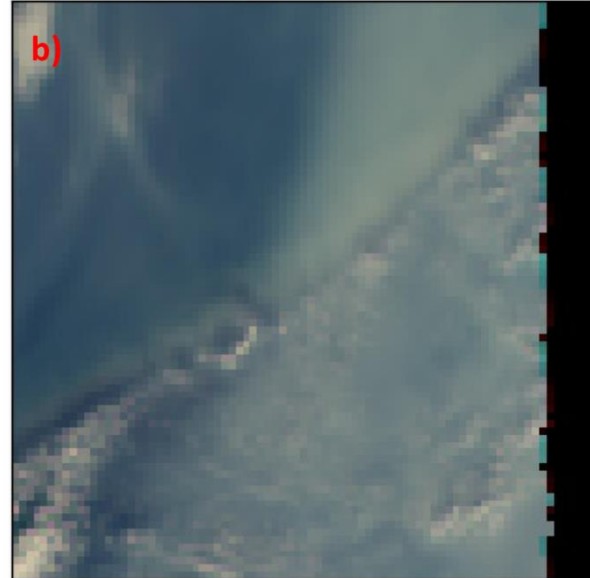

### MISR RA Rrs RGB

### MISR RA Aerosol Optical Depth (558 nm)

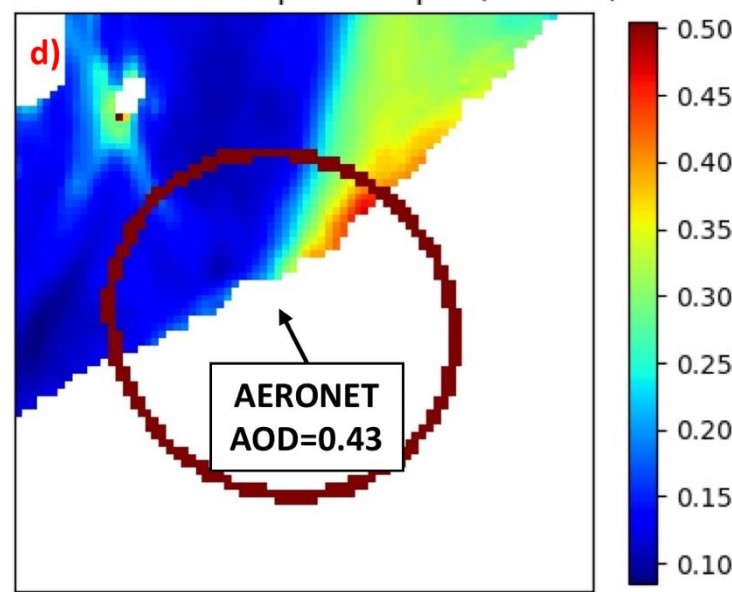

AERONET AOD=0.43

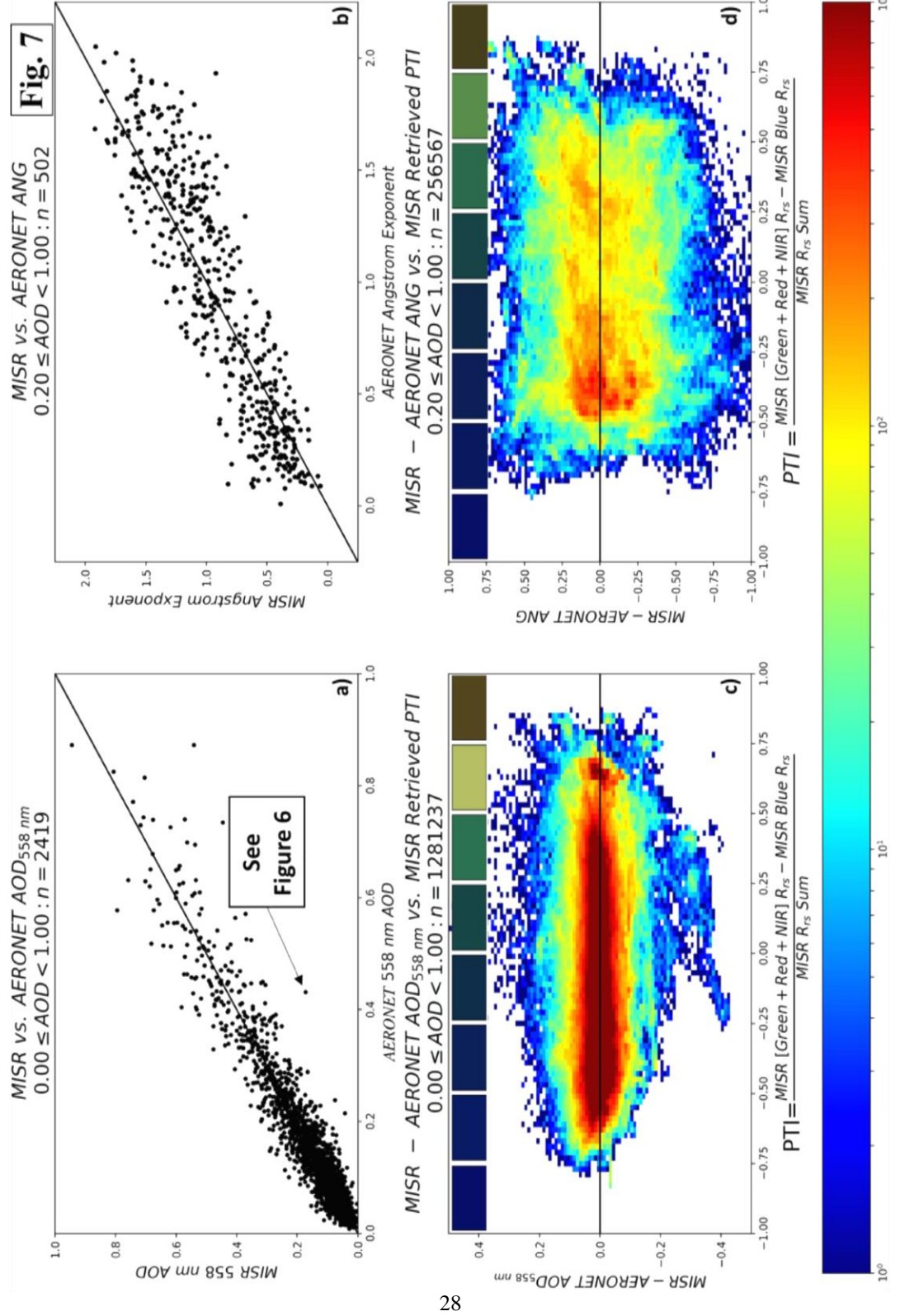

