# Peer review of "Updated MISR Over-Water Research Aerosol Retrieval Algorithm Part 2: A Multi-Angle Aerosol Retrieval Algorithm for Shallow, Turbid, Oligotrophic, and Eutrophic Waters"

_Atmospheric Measurement Techniques, 2018_

## Referee Comment (RC1) · Anonymous Referee #3 · 7 Sep 2018

The Authors present an approach to derive atmospheric and ocean color products using MISR in optically complex environments. Overall, I found the manuscript to be well written and compelling. The work builds on the Authors' previous studies, but in a meaningful way – that is, it provides a solid next step and advances the field. This manuscript meets the significance and scientific / presentation quality criteria to be published in the AMT discussion forum.

[Figure]

Several minor issues caught my eye while reading the manuscript.

Specific comments:

Page 2, Lines 1-3: Recommend adding a transition statement between the first two sentences. Not all readers will understand the second and third sentences without some context regarding the use of ocean color satellite instruments to study ocean productivity.

Page 2, Line 11: "local" 10:30 AM equator crossing time

Page 2, Lines 18-19: How do cameras with different viewing geometries all maintain a 275 m pixel size (I would expect nadir = 275 m, but off nadir > 275 m).

Page 2, Lines 27-28: "can be directly measured in the field by pointing a spectro-radiometer (or sun-photometer) at the ocean and dividing by the measured incident irradiance" is not correct. Deriving Rrs from above-water instrumentation also requires measuring sky radiance in the same plane as water-leaving radiance, plus making an estimate of surface reflection. The statement is true for in-water measurements that profile from depth to the surface to derive upwelling radiance.

Page 2, Lines 30-31: FWIW, the classic and most widely recognized reference for empirical ocean color algorithms is O'Reilly et al. 1998 in J. Geophys. Res. Oceans.

Page 2, Lines 31-32: FWIW, the only references in this sentence the appropriately support the statement are Maritorena 2002 and Werdell 2013 (e.g., Lee 2002 does not retrieve Chl).

Page 4, Line 24: Any sense of the sensitivity of the retrievals that results from use of a fixed surface pressure vs. a dynamic one?

Page 6, Line 11: Is there a reference for this?

Figure 4 and caption: The circles represent in situ measurements from a hand-held sun photometer, not AERONET, correct? If not, please describe how the AERONET

instrumentation moved throughout the scene.

---

## Referee Comment (RC2) · Anonymous Referee #1 · 5 Oct 2018

Review of AMT-205-2018 by Limbacher and Kahn

Summary: The authors have upgraded their research algorithm (RA) in order to simultaneously derive aerosol optical depth (AOD), Ångström exponent (ANG) and remote sensing reflectance (Rrs) from MISR observations over water. AOD and ANG are validated (compared to AERONET) and retrievals of Rrs and gradients appear realistic. There are case studies over a variety of water types (ranging from clear to turbid -

based on a new productivity/turbidity index scale (PTI)), and validation of AOD and ANG are from more than 2000 retrieved cases.

Overall: This paper is a suitable for AMT and provides a nice scientific analysis. Overall, I have no major objections to this paper being published, and have no compelling need for additional analyses. The results look good, and I would like to see (in a future study), the algorithm going "global". The figures are informative, and almost works of art (they will make beautiful posters).

I am curious why there are no MODIS-OC or AERONET co-authors, (or MISR-SA authors for that matter), considering the heavy use of their products. Yes, the products are freely available, but I don't believe acknowledgment-only is sufficient in this case.

One gripe is the appearance of too much self-referencing within the text. I believe the overall citation list is fair, but we don't need to see so much "Limbacher and Kahn" and "Kahn et al." within the text. I would rather see this space used to quickly describe the things that may be within these references. For example, I find lines 1-15 on Page 4 to be very confusing. I also don't like the equations. Is there any way to write these things without so many subscripts and superscripts? Finally, there are places in the text where the authors are claiming that because it looks good, it must be good (page 9, line 16 for example).

I wonder about the PTI metric. It appears useful, but calling it "productivity" may need validation. I do not have a better suggestion for the name, but should be more descriptive of what it is, rather than what is the desired inference.

I tried something new, which is to annotate the PDF (with red text). Some of my comments are questions to answer; others are possible suggestions for text revision. Others are tiny issues of punctuation. I hope the uploaded supplement will solve the problem of trying to align line/page numbers when revising the text.

Overall, a good effort, and I think publishable with minor revisions.

[Figure]

Please also note the supplement to this comment:
https://www.atmos-meas-tech-discuss.net/amt-2018-205/amt-2018-205-RC2-supplement.pdf

―――――――――――――――――――

**Supplement:**

[revised manuscript text omitted]

**Not sure how focal lengths and 15 years yields something sufficient for heteoreous coastal waters.**

In recent years, many in-situ measurements of near-surface Chlorophyll-a concentration (*Chl*, and other biological

**Don't understand "in concert". How about taken simultaneously, which offers a way to empirically go from optics to concentrations?**

constituents) have been taken in concert with measurements of remote-sensing reflectance ($R_{rs}$; sr$^{-1}$). $R_{rs}$ represents the *normalized* (see *Morel et al.*, 2002) upward directed radiance ($L_w$; W m$^{-2}$ µm$^{-1}$ sr$^{-1}$) just above the ocean surface, divided by the bottom-of-atmosphere, downward-directed spectral irradiance ($E_d$; W m$^{-2}$ µm$^{-1}$). $R_{rs}$ is widely used in the ocean color community because it can be directly related to near-surface biological proxies (measured in-situ; such as Chlorophyll-a), and can be estimated in the field from surface remote-sensing observations, as described in *Mobley* [1999]. As much remote-sensing-based ocean color research has focused on relating measurements of biological proxies to $R_{rs}$ empirically (*Werdell and Bailey*, 2002; 2005), two different strategies have emerged that make use of $R_{rs}$. The purely empirical approach to retrieving ocean color properties relates ratios of normalized $R_{rs}$ to parameters such as *Chl* statistically (e.g. see *O'Reilly et al., 1998, Morel and Gentili*, 2009, *Hu et al.*, 2012). A semi-analytical method involves using a combination of

radiative transfer theory (*Morel et al.*, 2002) and empirical observations (e.g. *Morel and Prieur*, 1977, *Lee et al.*, 2015) to retrieve *Chl* and other parameters (e.g. *Maritorena et al.*, *2002*, *Werdell et al.*, 2013). Fundamentally, both techniques rely on high-quality retrievals of $R_{rs}$, which is dependent on the quality and spectral dependence of the atmospheric correction (*Kahn et al.*, 2016). **And the quality of the measurements of TOA radiance/reflectance**

*Mobley et al.* [2016]  summarize the atmospheric correction procedure implemented in the ocean color algorithm used for MODerate resolution Imaging Spectroradiometer (MODIS) and SeaWiFS at NASA Goddard **Space Flight Center**. For dark (Case I) water retrievals, where the remote-sensing reflectance is negligible for the near-infrared wavelengths (NIR) **define SeaWIFS, please**, the retrieval process follows directly from the one proposed by *Gordon and Wang* [1994], with updates made to the aerosol optical models (*Ahmad et al.*, 2010) and spectral bands used. For Case II waters, where $R_{rs}$ in two NIR bands used is not necessarily negligible (or even identical), the algorithm follows an iterative scheme described in *Bailey et al.* [2010], initialized with a guess of $R_{rs}(NIR)=0$.

With nine cameras taking measurements at four wavelengths, MISR has the angular information content needed to retrieve information about aerosol properties, even over bright turbid water, but lacks the specific spectral bands for ocean color applications, though it can fill in glint-contaminated regions of single-view instruments such as MODIS (*Limbacher and Kahn*, 2017). As such, this paper focuses on MISR's ability to retrieve the atmospheric component, aerosol amount and type, over shallow, turbid, and eutrophic waters, as well as oligotrophic waters. The paper is organized as follows: section 2 outlines the datasets and methodologies used, example retrievals are shown in section 3, validation is presented in section 4, and conclusions are given in section 5.

**2 Upgraded MISR Research Aerosol Retrieval Algorithm (RA) and Comparison datasets**

**2.1 The Over-Water MISR RA Methodology**

**Let's combine 1st two sentences and make sound positive: "Unlike MISR standard aerosol (SA) product which provides aerosol information at 4.4 km globally, the research algorithm (RA) provides detailed aerosol parameters on a case-by-case basis at 1.1 km or 275 m resolution. These details include…**

**Note SA now provides at 4.4 km resolution, so newer references?**
Unlike the MISR standard aerosol product (SA; *Diner et al.*, 2008; *Martonchik et al.*, 2009), which provides publicly accessible aerosol amount and type information globally (*Kahn et al.*, 2010, *Kahn and Gaitley*, 2015; *Garay et al.*, 2017), the

[revised manuscript text omitted]

*Some how this text is really clunky. How about start by saying. To retrieve the Rrs signal from the TOA reflectance, we need to account for the roughened surface, the molecular scattering, and the geometry of observation. For roughened surface, we… For Rayleigh…*

**2.1.1 Research Algorithm Detailed Description**

*must account for*

*- function of surface windspeed*

Because we  the effects of a roughened ocean surface (glint and whitecaps) in order to retrieve the remote-sensing

*what is CCMP?*

reflectance, we use cross-calibrated multi-platform (CCMP; 0.25°, 6 hourly) version 2 data (*Atlas et al.*, 2011, *Wentz et al.*,

*why 10 m wind and not surface (2 m) wind?*

2015) to prescribe the 10-meter wind speed. Surface pressure (that determines the Rayleigh scattering contribution to TOA reflectance) is prescribed as 1013.25 mb, as this allows us to remove one dimension from our look-up-table (LUT), and has minimal impact over ocean, including coastal regions (though it might have an impact over elevated inland lakes). Once the LUT has been interpolated to the appropriate solar/viewing geometry and wind speed, we then iterate through our grid of AOD, calculating the $R_{rs,\lambda}$ needed to compute $M$. This is done by taking the derivative of (2) with respect to $R_{rs,\lambda}$, setting it equal to zero, and solving for $R_{rs,\lambda}$.

*The error of assuming a constant 1013 is minimal, except for over elevated inland lakes. Or is it? +/- 10 mb or 1% of atmosphere leads to 0.002 error at 446 wavelength.*

[revised manuscript text omitted]

Again, not be snarky, but just because the algorithm separated surface and atmosphere, does not mean it did it correctly. Instead say that the algorithm attempts to separate surface and atmosphere even under high aerosol loadings (AOD=0.5). Qualitatively, it seems reasonable becuase…

[Figure]

[Figure]

RA AOD and $R_{rs}$ agree well with each other, although MODIS suggests that the aerosol might **radius** be a bit smaller than the RA reports (panel 2b compared to 2f), and that the water might be a little less green (2c compared to 2g). Although the MISR RA AOD in panel 2a does not appear to be correlated with the water color seen in panel 2d, indicating successful surface-atmosphere separation, it does appear as though retrieved ANG might be biased a bit low in the turbid water regions (discussed further in section 4 below).  AOD and ANG retrieved from the MISR RA and MISR SA agree extremely well with each other, even though the comparisons are confined to the non-turbid water regions of the scene where the SA provides coverage.

Figure 3 shows results for the eutrophic northern Caspian Sea on September 19, 2015.  The RA (3b) indicates that a plume comprised of small (ANG ~1.5), spherical non-absorbing (shape and light-absorption retrieval plots not shown) aerosol is present over a ~20,000 km$^2$ region in the north-central part of the plotted domain.  Within the plume, the RA indicates that the mid-visible AOD varies between 0.10 and 0.15, whereas outside the plume the retrieved AOD is ~ 0.05.  This skill in identifying relatively low AOD plumes over bright water is due to the multiple view angles provided by MISR.  Because the portion of TOA reflectance attributed to the retrieved $R_{rs}$

decreases with increasing view angle, the spatial pattern of the MISR Df (70˚ forward view angle)  NIR reflectance (panel 3k) correlates well with the retrieved AOD (panel 3a).  Although the MISR RA shows a great deal of skill separating the effects of atmospheric and oceanic scattering, Figures 3b and 3d indicate some surface artifacts might be aliasing the ANG retrieval.  That said, the artifacts still do not lead to substantially different AOD retrievals between the dark-water and eutrophic-water portions of the scene.  MODIS actually masks much of the eutrophic region for this scene, with large discrepancies found between the MISR RA and MODIS for much of the remaining portions.  The MISR SA does not report any aerosol retrievals over water in this region, probably due to shallow water masking.

*(margin note: skill? means it must be validated.)*

In Figures 4 and 5 we present results from the MISR RA, MODIS OC algorithm, and the MISR SA for the eastern

Yellow Sea region on March 13, 2012.  Note that although AERONET data are plotted for this region, results over land may not be representative of the air column over nearby ocean, particularly if the AERONET station is elevated.  Panels 4a and 4e show good AOD agreement between the MISR RA and MODIS, although panels 4b and 4f indicate rather large discrepancies in ANG between the two retrievals.

Although AERONET data from the DRAGON deployment in S.E. Asia show quite a bit of variability in ANG

(~1.1-1.5), it does appear that MISR ANG is probably low-biased and MODIS ANG might be biased a bit high. Not surprisingly, this discrepancy between MISR and MODIS ANG shows up in the $R_{rs}$ ratios (panels 4c and 4g, respectively).  Note that in the southwest corner the of the scene, where the MISR RA retrieves AOD of up to ~1, the MISR RA is still capable of retrieving water color (panels 4d and 4l), whereas MODIS does not provide results

*(margin note: "quite a bit", "appear", "probably", "might be". Remove these qualifiers to improve the sentence.)*

[Figure]

for this part of the scene. In Figure 5, we present the MISR-retrieved $R_{rs}$ RGB for this scene juxtaposed with a Rayleigh-corrected MISR nadir true-color image and the MODIS retrieved $R_{rs}$ RGB.

Additional examples are included in Supplemental Material: a biological bloom in the East Argentine Sea, and three
cases over the Bohai Sea along coastal China east of Beijing, showing results for low, moderate, and high AOD loading.

**4 Validation of the MISR Over-Water RA AOD and ANG Using AERONET**

This sentence reads as if the L&K 2017 paper also compared against V23 4.4. I think you mean to say that here, MISR RA at 1.1 km is compared with 17.6 SA (V22) as well as new 4.4 SA (V23)

As in *Limbacher and Kahn* [2017], MISR RA aerosol retrievals are performed at 1.1 km resolution, compared with the 17.6 km for the versions of the RA used in *Limbacher and Kahn* [2014; 2015], as well as the new version 23 MISR SA 4.4 km
product (*Garay et al.*, 2017). For validation against AERONET, we average all good-quality MISR 1.1 km retrievals surrounding the AERONET site, yielding one value per coincidence. We consider AERONET/MISR coincidences only if the following conditions are met:

- AERONET temporal variability (max-min for all 4 MISR bands) $< 0.05 + 0.1*AOD$.   Over what time intervalwh?
- AERONET reported elevation <100 meters. This is necessary for two reasons:
- AERONET AOD (and ANG) might not be comparable to our over-water retrievals (ocean retrievals are all at sea-level).
    - The LUT we use (for water retrievals only) contains only one surface pressure value (1013.25 mb), meaning that retrievals over elevated inland lakes could suffer from non-negligible errors (depending on elevation).
- At least one AERONET observation on each side of the temporal averaging window (±30 minutes of MISR overpass).
- At least 5% of MISR observations within 25 km of the AERONET site result in good-quality MISR over-water aerosol retrievals. A good-quality MISR aerosol retrieval requires **all** the following pixel-level criteria to be met (most of which are for cloud screening):
- MISR RA cost function ($M$) $< 1$, which indicates a good model fit to the observations.
    - MISR RA maximum channel-specific cost function $< 0.5$, to screen out clouds that might only be visible in one or two cameras.
    - MISR RA $M/M'' < 10^{-3}$, as this ratio tends to increase when clouds are present.
    - Additionally, to improve cloud detection, we flag all MISR retrievals immediately surrounding any pixel
whose aerosol retrieval does not meet the three quality thresholds outlined above.

These constraints yield 2,419 MISR-AERONET coincidences for the four years of MISR data we currently have processed (four years interspersed between September 2000 and November 2016).

[Figure]

Results from the statistical comparison of MISR and AERONET are shown in Tables 1 and 2 (558 nm AOD and ANG, respectively). In order to identify co-variation between retrieved surface albedo and aerosol properties, we perform the following comparisons for both AOD and ANG against AERONET:

**why arrows?**

- Average→ Average all good-quality MISR aerosol retrievals within 25 km of the AERONET site,

- Lowest 10%→ Average only those good-quality MISR aerosol retrievals where the retrieved 558 nm (green) $R_{rs}$ is lower than the 10th percentile value for that specific scene,

- Highest 10%→ Average only those good-quality MISR aerosol retrievals where the retrieved 558 nm (green) $R_{rs}$ is higher than the 90th percentile value for that specific scene.

Figure 6 illustrates how the data is parsed and averaged; it provides some context for the AOD and ANG scatterplots presented in Figure 7, which shows both scene-averaged values for ANG and AOD in addition to 2-d histograms of AOD and ANG errors as a function of water color. We create the following productivity/turbidity index (PTI) that allows us to characterize MISR retrieval errors against a single water color parameter (similar to NDVI over land):

$$PTI = \frac{MISR\ [Green+Red+NIR]\ R_{rs} - MISR\ Blue\ R_{rs}}{MISR\ R_{rs}\ Spectral\ Sum}. \qquad (8)$$

**I don't understand this equation. How does one add MISR (Green + Red + NIR). And what is "Spectral Sum"? Finally, are you sure it is "productivity"?**

**How do you know this is a cloud?**

[revised manuscript text omitted]

---

## Author Comment (AC1)

**Reviewer #3**
**We thank reviewer 3 for their suggestions and feedback.**

The Authors present an approach to derive atmospheric and ocean color products using MISR in optically complex environments. Overall, I found the manuscript to be well written and compelling. The work builds on the Authors' previous studies, but in a meaningful way – that is, it provides a solid next step and advances the field. This manuscript meets the significance and scientific / presentation quality criteria to be published in the AMT discussion forum.

Several minor issues caught my eye while reading the manuscript. While not required for this initial review, these issues are provided below in case the Authors wish to get feedback.

Specific comments:

Page 2, Lines 1-3: Recommend adding a transition statement between the first two sentences. Not all readers will understand the second and third sentences without some context regarding the use of ocean color satellite instruments to study ocean productivity.
**We don't quite understand what the reviewer means by this.**

Page 2, Line 11: "local" 10:30 AM equator crossing time
**Added, thank you.**

Page 2, Lines 18-19: How do cameras with different viewing geometries all maintain a 275 m pixel size (I would expect nadir = 275 m, but off nadir > 275 m).
**Added that MISR uses different focal lengths for each camera.**

Page 2, Lines 27-28: "can be directly measured in the field by pointing a spectroradiometer (or sun-photometer) at the ocean and dividing by the measured incident irradiance" is not correct. Deriving Rrs from above-water instrumentation also requires measuring sky radiance in the same plane as water-leaving radiance, plus making an estimate of surface reflection. The statement is true for in-water measurements that profile from depth to the surface to derive upwelling radiance.
**Changed the statement to read: "$R_{rs}$ is widely used in the ocean color community because it can be directly related to near-surface biological proxies (measured in-situ; such as Chlorophyll-a), and can be estimated in the field from surface remote-sensing observations as described in *Mobley* [1999]."**

Page 2, Lines 30-31: FWIW, the classic and most widely recognized reference for empirical ocean color algorithms is O'Reilly et al. 1998 in J. Geophys. Res. Oceans.
**Thank you, added.**

Page 2, Lines 31-32: FWIW, the only references in this sentence the appropriately support the statement

are Maritorena 2002 and Werdell 2013 (e.g., Lee 2002 does not retrieve Chl).
**This was meant to be understood as Lee 2002 retrieving "other" parameters, but we have removed the citation to prevent confusion.**

Page 4, Line 24: Any sense of the sensitivity of the retrievals that results from use of a fixed surface pressure vs. a dynamic one?
**Over ocean we would expect negligible impacts on retrieved aerosol, over elevated inland lakes, there probably will be an impact.**

Page 6, Line 11: Is there a reference for this?
**I don't think there is a great estimate of *true* measurement uncertainty out there, although there are plenty of educated guesses.**

Figure 4 and caption: The circles represent in situ measurements from a hand-held sun photometer, not AERONET, correct? If not, please describe how the AERONET instrumentation moved throughout the scene.
**We have added that these circles represent AERONET retrievals during the southeast Asia leg of the DRAGON campaign (many AERONET CIMEL sun-photometers, not hand-held ones).**

---

## Author Comment (AC2)

**Reviewer #1**
**We thank reviewer 1 for their suggestions and feedback.**

Summary: The authors have upgraded their research algorithm (RA) in order to simultaneously derive aerosol optical depth (AOD), Ångström exponent (ANG) and remote sensing reflectance (Rrs) from MISR observations over water. AOD and ANG are validated (compared to AERONET) and retrievals of Rrs and gradients appear realistic. There are case studies over a variety of water types (ranging from clear to turbid - based on a new productivity/turbidity index scale (PTI)), and validation of AOD and ANG are from more than 2000 retrieved cases.

Overall: This paper is a suitable for AMT and provides a nice scientific analysis. Overall, I have no major objections to this paper being published, and have no compelling need for additional analyses. The results look good, and I would like to see (in a future study), the algorithm going "global". The figures are informative, and almost works of art (they will make beautiful posters).

I am curious why there are no MODIS-OC or AERONET co-authors, (or MISR-SA authors for that matter), considering the heavy use of their products. Yes, the products are freely available, but I don't believe acknowledgment-only is sufficient in this case.

**Although we appreciate the reviewer's perspective on this, the authors conceived the underlying ideas, developed the algorithm, wrote the text, and performed all the analysis. All products used here are freely available, except for the MISR RA, which is the authors' algorithm.**

One gripe is the appearance of too much self-referencing within the text. I believe the overall citation list is fair, but we don't need to see so much "Limbacher and Kahn" and "Kahn et al." within the text. I would rather see this space used to quickly describe the things that may be within these references. For example, I find lines 1-15 on Page 4 to be very confusing. I also don't like the equations. Is there any way to write these things without so many subscripts and superscripts? Finally, there are places in the text where the authors are claiming that because it looks good, it must be good (page 9, line 16 for example).

**This algorithm builds on prior work of the authors (this is also a "Part 2" paper), who have worked collaboratively for the last 8 years on this project. Many algorithm refinements that make the current advances possible are described in our previous papers, which is why there are many references to them. We have tried to consolidate citations, but are aware that others might question the justification for some steps if we didn't provide adequate references. We have also added that f/Q represents essentially the BRDF of the ocean's color and have tried to make the paragraph clearer in general.**

I wonder about the PTI metric. It appears useful, but calling it "productivity" may need validation. I do not have a better suggestion for the name, but should be more descriptive of what it is, rather than what is the desired inference.

**As water color relates to both productivity and turbidity, we think the calling it a productivity/turbidity index seems appropriate.**

I tried something new, which is to annotate the PDF (with red text). Some of my comments are questions to answer; others are possible suggestions for text revision. Others are tiny issues of punctuation. I hope the uploaded supplement will solve the problem of trying to align line/page numbers when revising the text.

**We have reviewed the annotated file, and made the indicated, minor edits to the text in many cases.**

Overall, a good effort, and I think publishable with minor revisions.

Specific comments in the text:

P1 L10:  Replace "the planet" with Earth
**Done**

P2 L2:  What do you mean by biologically productive, and suggest a reference?

**Chl-a maps from the MODIS OC website give an indication of biological productivity.  We don't think a reference is really necessary for such a generally accepted statement, but have added *Behernfeld et al.* (2005) as a reference.**

**Behrenfeld, M. J., E. Boss, D. A. Siegel, and D. M. Shea (2005), Carbon-based ocean productivity and phytoplankton physiology from space, Global Biogeochem. Cycles, 19, GB1006, doi:10.1029/2004GB002299.**

P2 L7:  Suggest: One reason is that if attempting to use satellites to observe coastal waters, the atmospheric contribution to the measured top-of-atmosphere (TOA) radiance is large.

**In this case, we prefer our wording, as it makes clear the specific algorithm limitations that we are overcoming with the new approach.**

P2 L12:  Originally intended as a 6-year mission,
**Done**

P2 L15:  +/-
**Done**

P2 L16:  optical path length
**Done**

P2 L17:  Why only sometimes?
**Removed sometimes.**

P2 L22:  Not sure how focal lengths and 15 years yields something sufficient for heterogeneous coastal waters.

**The varying focal length allows the instrument to make measurements at roughly the same spatial resolution for all nine cameras, allowing for retrievals in heterogeneous coastal waters (especially in the high-resolution mode).  The 15 years is a reference to the local-mode dataset of high-resolution MISR data.  We have emphasized these point in the revised text.**

P2 L25:  Don't understand "in concert". How about taken simultaneously, which offers a way to empirically go from optics to concentrations?
**In concert with is a synonym for jointly (We have changed it to "jointly.").**

P3 L4:  And the quality of the measurements of TOA radiance/reflectance
**Right**

P3 L6:  Remove nicely
**Done.**

P3 L7:  Define SeaWIFS please, and change NASA Goddard to NASA Goddard Space Flight Center.
**Done**

P3 L23:  Let's combine 1st two sentences and make sound positive: "Unlike MISR standard aerosol (SA) product which provides aerosol information at 4.4 km globally, the research algorithm (RA) provides detailed aerosol parameters on a case-by-case basis at 1.1 km or 275 m resolution. These details include…

**We think it is important for the current paper to make clear the many differences between the SA and the RA.  We have revised these sentences as follows:**

**The MISR standard aerosol product (SA; Diner et al., 2008; Martonchik et al., 2009) provides publicly accessible aerosol amount and type information globally (Kahn et al., 2010, Kahn and Gaitley, 2015; Garay et al., 2017). In contrast, the RA can only process MISR data for selected locations and times, on a case-by-case basis, but offers spatial resolution down to 1.1 km or 275 m pixel size, and advances in radiometric calibration critical for aerosol-type retrieval, improved surface representation, and the option of a greatly expanded aerosol optical model climatology (Limbacher and Kahn, 2014; 2015; 2017).**

P3 L23:  Note SA now provides at 4.4 km resolution, so newer references?
**We mention that fact, and the appropriate reference Garay et al., 2017) is included.**

P4 L3:  AU → Astronomical Units?
**Added, thanks**

P4 L7:  I am curious how much "correction" all of these things do?
**This is in our previous work, which we reference extensively to address such questions.**

P4 L11:  What is f/Q correction?
**We now explain that f/Q represents a non-Lambertian bi-directional surface modification.**

P4 L12:  Maybe if you aren't using it now, then don't even mention until maybe discussion/conclusions
**f/Q corrections are fundamental to many ocean-color algorithms, so we feel it needs to be mentioned here..**

P4 L13:  Now, we are talking your algorithm, or the f/Q algorithm?
**f/Q is a correction**

P4 L14:  normalized to what? Again, tell me what you are doing now, and not what you are doing later.
**We removed this.**

P4 L15-16:  Don't understand "aggregate the resulting AOD… corresponding to best-fitting AOD"
**We have revised the text to clarify this.**

P4 L21:  suggest using the form: "where w is, Unc is, and pmodel is.. "
**We don't see any advantage to the suggested rewording.**

P4 L25:  Not sure "irradiant" is the correct word here, and symbol "E" usually refers to irradiance.
**Our description is accurate, but we have added that it is analogous to the TOA reflectance normalization.**

P4 L30:  Why do you need all of these self citations? Using exponential weighting, rather than specific thresholds, avoids the possibility of arbitrarily excluding some mixtures.

We were pointing out what we had done in prior work, but we have now removed those references here at your request.

P5 L1-10: Somehow this text is really clunky. How about start by saying. To retrieve the Rrs signal from the TOA reflectance, we need to account for the roughened surface, the molecular scattering, and the geometry of observation. For roughened surface, we… For Rayleigh…

**We prefer varying the sentence structure to minimize monotony. We did revise these sentences as follows:**

**To retrieve the remote-sensing reflectance, we prescribe the wind-driven effects of a roughened ocean surface (glint and whitecaps), using cross-calibrated multi-platform (CCMP; 0.25°, 6 hourly) version 2 data (Atlas et al., 2011; Wentz et al., 2015) 10-meter wind speed. The Rayleigh scattering contribution to TOA reflectance is prescribed with a 1013.25 mb surface pressure. This allows us to remove one dimension from our look-up-table (LUT) and has minimal impact over ocean, including coastal regions (though it might have an impact over elevated inland lakes).**

P5 L2: Change prescribe to must account for

**We do prescribe it.**

P5 L2: Are whitecaps a function of surface windspeed?

**Yes, as described in our previous papers. This is clarified in the above revision.**

P5 L3: What is CCMP?

**As stated, CCMP stands for cross-calibrated multi-platform, but we have added that this is 6 hourly wind data fused from multiple satellite instruments and model data.**

P5 L4: Why 10 m wind and not surface (2 m) wind?

**The widely used roughened surface model we adopted is designed for 10 m wind input.**

P5 L5: The error of assuming a constant 1013 is minimal, except for over elevated inland lakes. Or is it? +/- 10 mb or 1% of atmosphere leads to 0.002 error at 446 wavelength.

**Even a deviation of 50 mb is still only 0.01 in AOD in the blue (and is less in other channels). We suspect that actual surface pressure will be biased slightly higher than 1013, as we don't do retrievals over clouds.**

P5 L11: Do you have a reference to suggest the normal "range" of surface?

**We do not, but from our own experience, it can vary from nearly black to > 30% in shallow waters with bright sand, as illustrated in the case studies in Section 3.**

P5 L13: What is that dimension? M becomes only a function of AOD, right?

**Correct**

P6 L3: I have never seen so many subscripts on a term!

**We tried to be clear about the variable dependencies. The subscripts just refer to geometry and wind speed variables, with perturbations.**

P7 L22: Need accents on Angstrom Exponent: Check other instances too.

**Thanks! Changed this throughout the paper.**

P8 L8: What is the raw product?

**For clarity, we changed this to "…raw' data product, which provides AOD with minimal cloud screening"**

P8 L17: Why not something else? Reference needed?

**One can actually see the sun-glint with the MISR An (nadir-viewing) camera, indicating that it would be a problem for MODIS as well.**

P8 L23: Do you mean "albedo" - why do you retrieve albedo?
**This should be R$_{rs}$, thanks.**

P8 L24-25: Just because the RA retrieves here doesn't mean it should…
**This is why we validate in all types of water**

P8 L30-31: Again, not be snarky, but just because the algorithm separated surface and atmosphere, does not mean it did it correctly. Instead say that the algorithm attempts to separate surface and atmosphere even under high aerosol loadings (AOD=0.5). Qualitatively, it seems reasonable because…
**We validate where we can, both statistically and for specific cases, and suggest that the errors will tend to be similar in locations where validation is lacking.  We don't claim the results are perfect.**

P9 L1: aerosol → aerosol radius
**We don't think that is necessary here.**

P9 L16: skill? Means it must be validated
**The human eye can give a qualitative indication of skill, but we have removed "a great deal of".**

P9 L29-30: "quite a bit", "appear", "probably", "might be". Remove these qualifiers to improve the sentence.
**Done**

P10 L8-10: This sentence reads as if the L&K 2017 paper also compared against V23 4.4. I think you mean to say that here, MISR RA at 1.1 km is compared with 17.6 SA (V22) as well as new 4.4 SA (V23)
**We agree that this was not worded well and have made changes.**

P10 L13: Over what time interval?
**Moved the bullet talking about temporal averaging up so it is first.**

P11 L 4-7: Why use arrows? (→)
**Changed to dashes.**

P11 L13: I don't understand this equation. How does one add MISR(Green + Red + NIR). And what is "Spectral Sum"? Finally, are you sure it is "productivity"?
**These are MISR R$_{rs}$ sums. Spectral sum would be Blue + Green + Red + NIR.  We have added words to the text to make this clear.  Figure 7 illustrates how this index reflects water surface productivity/turbidity.  In future work, the index can be further verified with in-situ (or retrieved) Chl observations (or retrievals).**

P11 L15: How do you know this is a cloud?
**Its texture, shape, and color in the visible imagery all look like cloud.**

P11 L21-22: This is a very good result!!
**Thank you.**

P11 L28-29: I think you mean "more turbid". Saying water is "productive" is something to validate.
**We are just putting results in perspective.  We say here only that we \*expect\* coastal waters to be productive for a couple of reasons, one being that nutrient run-off can lead to near coastal blooms.**

P13 L10-11:  This is a great next step!
**Thanks.**